# Stacking sequence and interlayer coupling in few-layer graphene revealed by *in situ* imaging

Zhu-Jun Wang[1], Jichen Dong[2], Yi Cui[3], Gyula Eres[4], Olaf Timpe[1], Qiang Fu[5], Feng Ding[2], R. Schloegl[1] & Marc-Georg Willinger[1]

In the transition from graphene to graphite, the addition of each individual graphene layer modifies the electronic structure and produces a different material with unique properties. Controlled growth of few-layer graphene is therefore of fundamental interest and will provide access to materials with engineered electronic structure. Here we combine isothermal growth and etching experiments with *in situ* scanning electron microscopy to reveal the stacking sequence and interlayer coupling strength in few-layer graphene. The observed layer-dependent etching rates reveal the relative strength of the graphene–graphene and graphene–substrate interaction and the resulting mode of adlayer growth. Scanning tunnelling microscopy and density functional theory calculations confirm a strong coupling between graphene edge atoms and platinum. Simulated etching confirms that etching can be viewed as reversed growth. This work demonstrates that real-time imaging under controlled atmosphere is a powerful method for designing synthesis protocols for $sp^2$ carbon nanostructures in between graphene and graphite.

[1] Department of Inorganic Chemistry, Fritz Haber Institute of the Max Planck Society, Berlin-Dahlem D-14195, Germany. [2] Institute of Textiles and Clothing, Hong Kong Polytechnic University, Hong Kong 999077, China. [3] Vacuum Interconnected Nanotech Workstation, Suzhou Institute of Nano-Tech and Nano-Bionics, Chinese Academy of Sciences, Suzhou 215123, China. [4] Materials Science and Technology Division, Oak Ridge National Laboratory, Oak Ridge, Tennessee 37831, USA. [5] State Key Laboratory of Catalysis, Dalian Institute of Chemical Physics, Chinese Academy of Sciences, Dalian 116023, China. Correspondence and requests for materials should be addressed to M.-G.W. (email: willinger@fhi-berlin.mpg.de).

Engineering a bandgap without degrading electron mobility is the key to making graphene into a practical electronic material. The two main strategies for opening up a bandgap in graphene rely on size- and shape-dependent quantum confinement and charge transfer density modulation. In practice, various approaches, such as patterning of graphene nanoribbons[1–3], chemical doping or physisorption of various molecules[3,4], applying uniaxial tensile strain[5] or binding the graphene onto substrates[5,6], have been used to implement these strategies. However, they all compromise the intrinsic properties of graphene either by disturbing the $\pi$ electrons or by introducing boundaries and defects. In the case of hydrogen plasma etching, incomplete understanding of the underlying reaction mechanisms limits its application to a trial-and-error approach in which the production of well-defined graphene edge structures without disturbance of the basal plane remains an unsolved problem. A control of the electronic states in graphene is also possible by taking advantage of interlayer interactions. Indeed, ordered structures consisting of two or more layers of graphene represent a broad class of materials, where the electronic structure and properties uniquely change with each additional layer[7–10]. The coupling between graphene layers and their interaction with the substrate induces charge transfer density modulations. Although the cohesive interaction between graphene sheets is a relatively old topic that has been studied for more than 50 years[11–14], there remains a lack of experimental data regarding the effect of the substrate on the coupling between few-layer graphene (FLG). Bi-layer graphene (BLG) is of great technological interest because the presence of the second layer creates a semiconductor with a bandgap that can be tuned by gating. In contrast, tri-layer graphene (TLG) is a semimetal where gating can be used to change the conductivity. In principle, FLG allows maintaining high electron mobility with only minimal disturbance of the $\pi$ electron dispersion and without the formation of new boundaries[7–10]. The growth of BLG and FLG has been achieved on a variety of metal surfaces including Ni, Ni-Cu alloy, Cu, Ru, Ir and Pt catalysts[15–20]. The electronic properties of the FLG structures vary as a function of interlayer spacing, twist angle and stacking order. Well-known examples of stable staking orders (polytypes) that have distinct electronic properties are the Bernal (AB) and the rhombohedral (ABC) stacking. The formation of a particular stacking order is known to be strongly influenced by the synthesis method and substrate type[21,22]. Specifically, the stacking order can be affected by the vertical stacking sequence of adlayer graphene (ALG). The two vertical stacking sequences in FLG are generally discriminated in the graphene literature as wedding cake (WC) and inverted WC (IWC) models, indicating that the ALG forms either above or inserts below an already grown layer[23]. For substrates that are characterized by a low carbon solubility and weak graphene–substrate interaction, such as Cu, the stacking sequence of ALG was confirmed to be IWC by isotope labelling and Raman measurements[24]. However, for catalysts that are characterized by higher carbon solubility than Cu such as Ir, Pt, Rh, Ni, Co and Ru, the stacking sequence is harder to determine unambiguously. Indeed, the ALG can form either by surface in-plane feeding or by carbon segregation from the bulk of the substrate during cooling[20]. In the case of Ru, Sutter et al.[19] described the stacking order by the WC model on the basis of combined in situ low-energy electron microscopy and charge transport measurements. In contrast, Sun et al. using post-growth scanning electron microscopy (SEM) imaging found that multilayer growth on Pt is dominated by carbon precipitation below already formed layers[25]. Real-time imaging is a powerful tool for studying growth kinetics because it enables extracting quantitative data from the changes of the shape and size of graphene islands during their evolution in response to externally

controlled environments. In a recent report we demonstrated the effectiveness of in situ environmental SEM (ESEM) for studying the mechanistic details of graphene chemical vapour deposition (CVD) on Cu (ref. 26).

In the present paper we take real-time imaging one step further by monitoring isothermal etching of graphene layers on polycrystalline Pt foils to probe the interlayer coupling and reveal the stacking sequence in FLG. We show that etching rates are proportional to the relative coupling strength and that the interaction between two neighbouring graphene layers is significantly weaker than the interaction of SLG with the Pt surface. The observed anisotropic etching behaviour is analysed and related to the interaction of graphene edge atoms with Pt step edges. Finally, the interpretation of the dynamic data is complemented by post-growth characterization using micro-Raman spectroscopy, scanning probe microscopies (atomic force microscopy (AFM) and scanning tunnelling microscopy (STM)), high-resolution transmission electron microscopy and theoretical calculations. The ability to probe the interlayer interactions in graphene is important for developing key processing steps such as selecting the ideal substrate for facilitating SLG transfer and the tuning of the properties of FLG by controlling the sequencing of ALG stacking and the number of layers. The broader significance of this work is in demonstrating that etching in combination with direct imaging of in-plane dynamics in response to well-controlled experimental environments is a facile approach for deriving information about interlayer coupling that governs the vertical stacking behaviour of two-dimensional materials.

## Results

**Growth and characterization of FLG.** The growth and etching of graphene were both performed in the chamber of an ESEM (see Methods section). The ESEM enables real-time imaging of the shape and size evolution of graphene islands in relation to the Pt grain structure and surface features. Observations can be performed as a function of the background atmosphere and temperature during both, growth and etching of graphene[26]. FLG was grown by isothermal CVD using ethene ($C_2H_4$). After initial growth of FLG islands in a $C_2H_4/H_2$ atmosphere, the $C_2H_4$ flow was turned off to perform isothermal etching in pure $H_2$ at a total pressure of 25 Pa (for experimental details see Methods section). We attribute the etching to carbon bond breaking by atomic hydrogen that is produced by dissociation of $H_2$ on a Pt surface that is known to be a highly efficient catalyst for promoting $H_2$ dissociation[27]. Post-growth characterization by Raman, STM and high-resolution transmission electron microscopy was used to confirm the high quality of the investigated graphene (Supplementary Fig. 1).

The key for probing the dynamics of ALG during both growth and etching is the ability to differentiate individual layers stacked on top of each other (Supplementary Movie 1). A wide contrast range is provided by the secondary electron signal, which is sensitive to changes in the surface charge state, electronic structure, work function and variations in secondary electron yield[28]. The in situ SEM image in Fig. 1a and the plot in Fig. 1b illustrate the stepwise variation of the contrast that allows identification of up to nine individual graphene layers, starting with the brightest first layer in contact with the substrate[29]. In addition, the in situ SEM images of edge misalignment between mutual layers and individual sheets provide real-time information on the evolution of the rotation angle between growing layers, and formation of the stacking order[30]. Figure 1c illustrates a 30°–30° rotation between successive layers and Fig. 1d shows ABA or ABC stacked graphene with a hexagonal shape distorted by strong interaction with Pt step edges on the left.

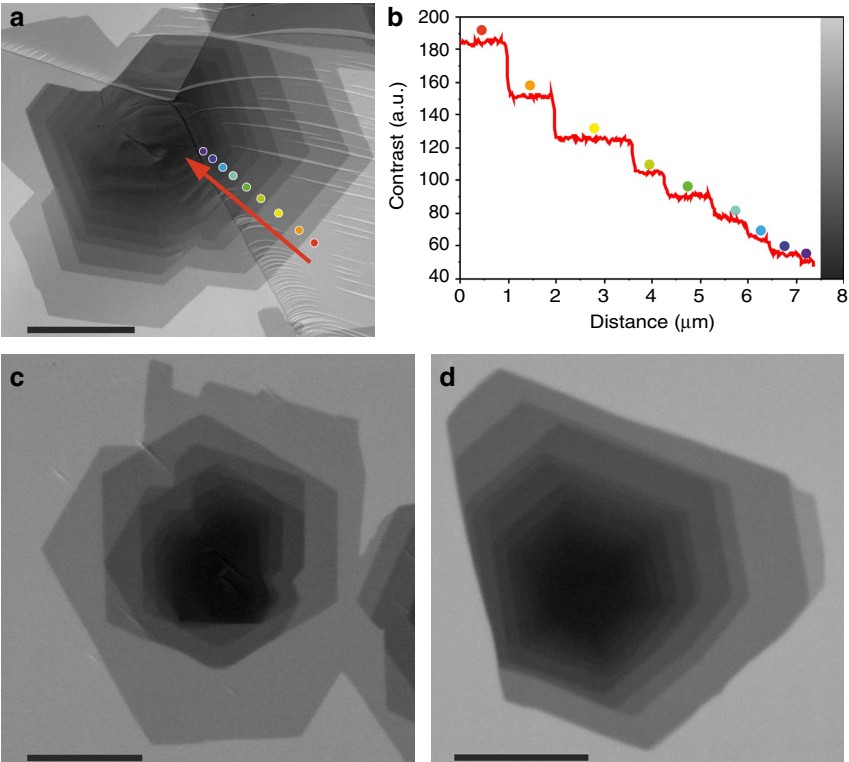

**Figure 1 | Real-time characterization of graphene sheets. (a,b)** The ESEM image (**a**) shows a FLG stack. Along the red arrow, the brightness in the secondary electron image changes with each additional layer. (**b**) Line plot showing the change in contrast along the red arrow in (**a**). Up to nine layers can be distinguished within the contrast range. The different coloured dots along the arrow are intended to assist the assignment between layer number and grey value. Note that the lightest shade marked by a red dot corresponds to Pt covered by SLG. (**c**) Vertical layer stacking showing a 30° rotation between successive layers. (**d**) Hexagonal shape distorted by interaction with the Pt surface in ABA or ABC stacked FLG. Scale bars, 5 μm (**a**); 2 μm (**c,d**).

**Real-time characterization of FLG etching**. The shape evolution of single-layer graphene, BLG and TLG during hydrogen etching is illustrated in Fig. 2. Images 2a–d were recorded *in situ* at 900 °C in an atmosphere of 25 Pa hydrogen. Designated areas in Fig. 2a are replotted and compared directly in Fig. 2e. The time-dependent evolution of the shape and size of the topmost graphene layers in BLG and TLG during etching is illustrated by colour-coded plots and compared with etching of SLG in Fig. 2e. Individual shapes were extracted from frames of the *in situ* SEM movie provided as Supplementary Movie 2. It is important to note that the etching process is limited to the island edges, while the basal planes remain intact. Indeed, except for the very beginning of the etching process, where some holes appear at grain boundaries, no etching pits appear even after etching for more than 6,000 s (Fig. 2a–d). This behaviour implies that pure hydrogen etching is less aggressive and more controllable than plasma-assisted hydrogen etching, and that the graphene is of high quality. It can be seen that etching takes place simultaneously at the periphery of each individual layer, indicating that graphene edges in a vertical stack are equally exposed to the reactive hydrogen atmosphere. Also, the out diffusion of etching products from graphene edges in each layer to the surrounding atmosphere is not hindered by the presence of other layers. Thus, the etching behaviour indicates the absence of buried layers. Smaller sheets therefore grow on top of larger ones, indicating that isothermal CVD growth of FLG on Pt substrates follows the WC type stacking. During etching, the smallest topmost layer is the first to disappear in each stack. Hence, in TLG the third layer, and in BLG the second layer are first to disappear, illustrating that removal of layers can proceed in a layer-by-layer manner.

The evolution of the perimeter and area of the graphene islands during H$_2$ etching at 900 °C is plotted in Fig. 2f,g. The excellent linear fit for the perimeter and corresponding quadratic fit for the area is consistent with a detachment-limited etching process following first-order kinetics[31]. The line slopes in Fig. 2f correspond to the averaged radial etching rates of the respective layers in a. They are − 5.84 nm s$^{-1}$ for the first, − 16.01 nm s$^{-1}$ for the second and − 11.76 nm s$^{-1}$ for the third layer, and are thus different for different layers. With respect to the first layer, the etching speed of the second layer is higher by a factor of 2.74 and the one of the third layers by a factor of about 2. This implies different graphene-edge configurations and indicates a WC-like stacking. Indeed, in a WC configuration, graphene edge atoms are in direct contact with the Pt substrate only in the case of the first layer, while edge atoms of adlayers are located on top of a graphene sheet and are most likely hydrogen-terminated.

Under the assumption that the removal of carbon atoms from the perimeter of a graphene sheet can be described by an Arrhenius-type rate $r \sim \exp(- E_a/k_B T)$, it should be possible to estimate the relative strength of the graphene interlayer coupling.

$$r \sim \exp\left(- \frac{E_a}{k_B T}\right)$$
$$= \exp\left(- \frac{E_{C-C} + E_{Coupling} + \textit{unspecified contributions}}{k_B T}\right) \quad (1)$$

The apparent activation energy $E_a$ that is required for removing an edge atom by hydrogen etching contains several contributions. Amongst them, in-plane carbon–carbon bond breaking ($E_{C-C}$)

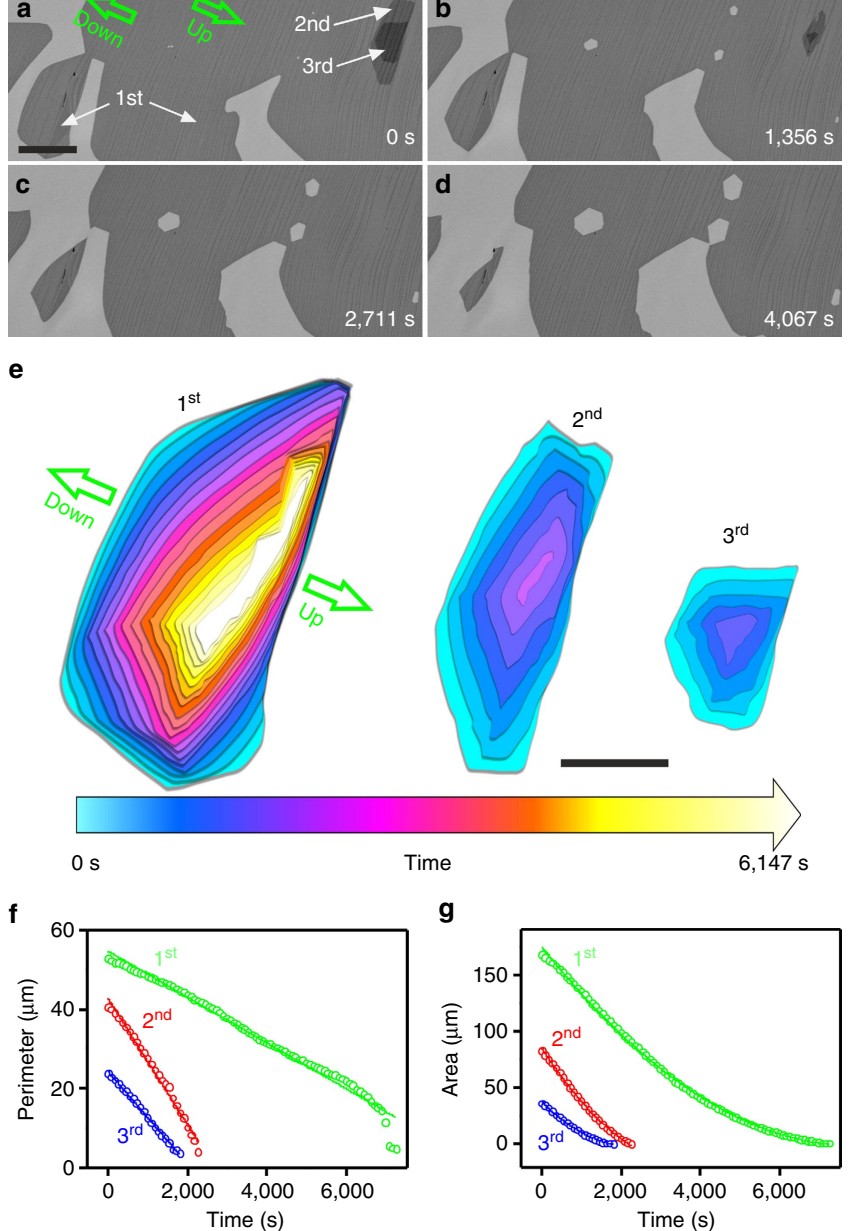

**Figure 2 | Evolution of isothermal CVD grown graphene layers during H$_2$ etching.** (**a**–**d**) Time-lapse image series showing the etching of SLG and the topmost layers in BLG and TLG. (**e**) Shape evolution of the respective layers during etching, reproduced as colour-coded superposition of outlines that were abstracted from images recorded at 3,600 s intervals (Supplementary Movie 2). (**f**) Evolution of the perimeters of the first, second and third layers in **a**, with corresponding linear fits. (**g**) Evolution of the area of the first, second and third layers in **a**, with corresponding quadratic fits. The green arrows in **a**,**e** indicate the up-step and down-step directions of Pt terraces. The scale bars in **a**,**e** measure 10 and 5 μm, respectively.

contributes the largest portion. However, there are also small contributions due to interlayer interactions between edge atoms and the graphene sheet underneath ($E_{Coupling}$). Since edge atoms in the second and third layers are located on top of a graphene sheet in the WC configuration, they face a similar local environment and are exposed to the same hydrogen-rich atmosphere during etching. Thus, the product of the etching process should be the same for both layers. Differences in the activation energy might therefore give a hint on differences in the interlayer coupling strength. By forming the ratio between the experimental etching rates of the second and third layers using equation (1), identical contributions to the activation energy, which are related to in-plane carbon–carbon bond breaking ($E_{C-C}$ + *unspecified contributions*) should

cancel out, leaving only terms due to different interlayer coupling:

$$E_{(Coupling-third-second)} - E_{(Coupling-second-first)} = k_B T \cdot \ln\left(\frac{r_{second}}{r_{third}}\right)$$
(2)

Using the experimentally determined etching rates for the second and third layers that are provided in the diagram of Fig. 2f in equation (2), we arrive at the estimation that the coupling of second layer edge atoms to the first layer is about 31 meV weaker than the one between third layer edge atoms and the second layer. Since this value represents the difference in graphene interlayer interaction probed by edge atoms, it cannot directly be translated

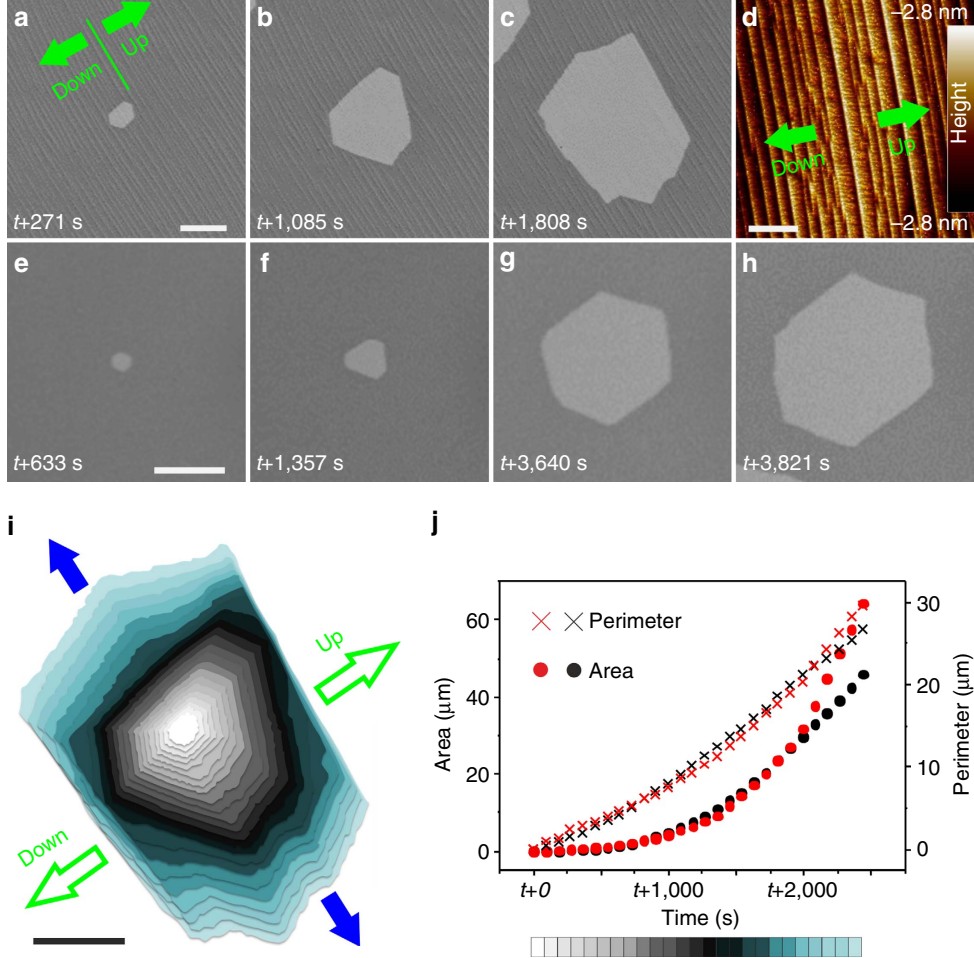

**Figure 3 | The anisotropic etching of vacancy islands.** (**a–c**) *In situ* SEM images recorded at 900 °C during H$_2$ etching showing the evolution of etch pits on a faceted Pt surface. *t* corresponds to the start time of dosing H$_2$ into the chamber. (**d**) AFM image recorded from the same Pt grain imaged in **a,b**. The graphene covered surface is characterized by graphene-induced Pt step bunching and surface reconstruction. (**e–h**) *In situ* SEM images recorded at 900 °C during H$_2$ etching showing the evolution of a vacancy island on a flat Pt surface. (**i**) Time-dependent change of the size and shape of the vacancy island shown in **a–c**. The superimposed shapes were extracted from frames recorded at 180 s intervals. Green arrows indicate the up-/downward direction of steps. Blue arrows indicate the direction of elongation along the terraces. (**j**) Line plots showing the evolution of the perimeter and area as a function of etching time, black symbols correspond to etching on the stepped Pt surface and red symbols to the case of the flat Pt surface. The scale bars in **a,d,e** and **i** measure 2 µm, 200 nm, 5 µm and 2 µm, respectively.

to the cohesive energy between the graphene sheets. Indeed, compared with the interlayer cohesive energy that is reported for graphite ($\sim$ 52 meV)[32], the energy difference determined on the basis of the different etching rates is quite large. Additional etching experiments revealed that the etching rates depend on the etching temperature and are influenced by the surface structure of the Pt grain (Supplementary Figs 2 and 3, Supplementary Movie 3 as well as Supplementary Note 1). Hence, anisotropic etching due to irregularities in the morphology of the substrate and resulting anisotropy in the shape of the graphene adlayers should be taken into account. Abstraction of the etching rate based on an integral shrinking perimeter is thus not sufficient for an accurate evaluation of the coupling strength experienced by edge atoms. However, in the case of perfectly flat Pt grains and symmetric hexagonal flakes, etching experiments performed at different temperature should even deliver layer-dependent activation energies for etching. Here we refrain from attempting to provide accurate numbers. Instead, we concentrate on the fact that etching experiments performed at different temperatures and on different grains confirmed that the second layer always etches at higher rate than the third

layer, and that the difference is not related to the size of the etching layers.

The faster etching speed of the second layer compared with the third layer indicates that the van der Waals interaction between the second and first layer is weakened due to the interaction between the first layer and the Pt substrate. In the case of copper substrates, it has previously been shown that coupling between SLG and the substrate induces *n*-type doping of the graphene sheet, which can be detected by a corresponding shift in the position of the C$_{1s}$ peak in X-ray photoelectron spectra[33]. A similar *n*-type doping was also observed in the case of ruthenium substrates[34]. The observed weaker coupling between the second and first graphene layers that we observe here is thus attributed to a charge imbalance imposed by the strong coupling of the first layer to the Pt substrate. The slow etching of the first layer is thus a direct confirmation of a strong interaction between graphene edge atoms and the Pt substrate.

**Strong anisotropy revealed by vacancy island etching.** A closer scrutiny of Fig. 2e reveals that etching does not uniformly shrink

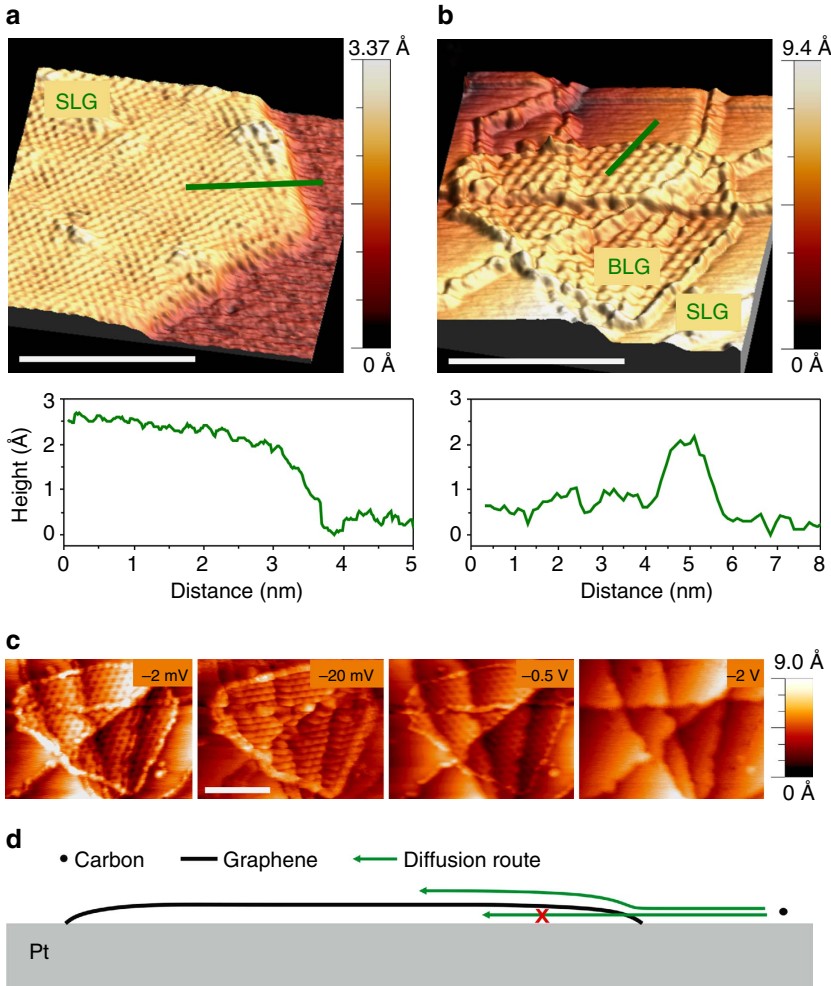

**Figure 4 | STM imaging of graphene edge structures.** STM images showing topographic contrast and corresponding height profiles along the green lines are shown in **a**,**b**. The images recorded at − 2 mV bias voltage and 1 nA tip current show a reduced signal at the edge of single-layer graphene that is in direct contact with Pt (**a**) and a higher signal due to increased electron density at edge atoms terminating the top layer in BLG (**b**). The dependence of the topologic contrast on tip voltage at a current of 1 nA is shown in **c** for the case of BLG. A schematic drawing of the bonding of graphene edges and its influence on the diffusion routes of carbon species is shown in **d**. Scale bars in **a**,**b**,**c** measure 4, 20 and 10 nm, respectively.

the islands. Instead, the islands become elongated in a direction along the Pt terraces that run perpendicular to the green up and down arrows. The anisotropy is most visible for the first graphene layer, which is in direct contact with the Pt substrate. This shape evolution indicates that the etching rates are slower in the direction perpendicular to the Pt steps than along the terrace. Asymmetric etching of islands is less pronounced for the second and third graphene layers, and is thus another indication of a strong interaction between graphene edge atoms and the Pt surface.

The interaction of graphene with Pt step edges is explored further by measurements of the shape evolution of vacancy islands or holes during isothermal etching of *in situ* grown SLG (Supplementary Movie 4). In Fig. 3a–c the size of the vacancy island increases starting after some finite time $t$ that is needed to open a small hole at a defect in the SLG. The shape evolution of the vacancy island during etching is plotted in Fig. 3i from frames recorded at 180 s intervals. The most distinctive feature of this hole is its highly anisotropic shape resulting from the transformation of a hexagon to an elongated polygon. Although atomic-scale surface features of the Pt substrate cannot be resolved by ESEM, we can clearly detect the larger steps that are formed by step bunching during growth[35]. Real-time imaging

during etching clearly demonstrates that anisotropy of vacancy islands is caused by the alignment of the etching front with Pt terrace edges, which are discernable as faint lines at roughly 45° in Fig. 3a–c. The step edges on the Pt surface are more visible in the AFM image that was recorded on the same Pt grain after the ESEM experiment (Fig. 3d). Similar to the islands, anisotropic etching of the hole in Fig. 3i proceeds by rapid elongation along the terraces marked by the blue arrows, while it is suppressed across edges by strong interaction with Pt atoms in the direction of the green arrows. In contrast, Fig. 3e–h and Supplementary Movie 5 show that if a hole forms on a large terrace devoid of steps, the hexagonal shape is preserved as the hole expands with time. The stability of the hexagonal shape indicates that the edges are zigzag terminated. A comparison of line plots in Fig. 3j reveals that the overall etch rates change as the shape of the hole evolves. The red symbols correspond to the perimeter and the area of uniformly expanding hexagonal holes on a terrace and thus, represent the intrinsic etching rates of graphene on Pt. The black symbols correspond to the expanding vacancy island on the stepped Pt surface. Initially, the black symbols overlap the red ones. But, they break away from the red ones at a point in time when elongation of the hexagonal shape in Fig. 3i sets in. This behaviour indicates that the detachment-limited removal of

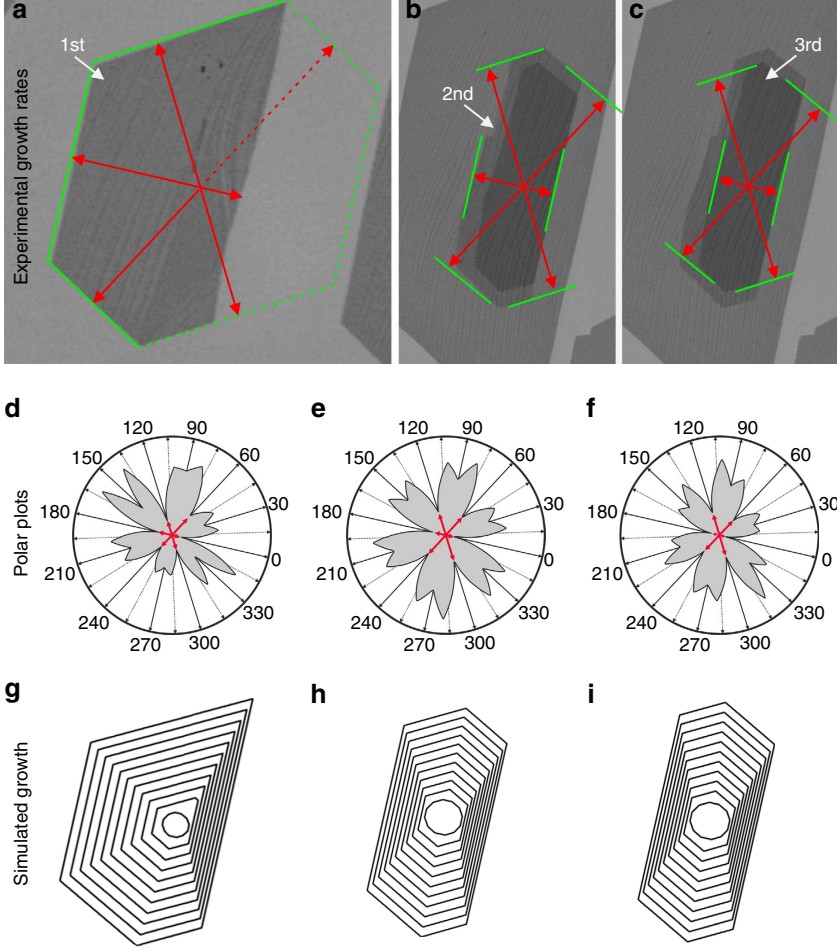

**Figure 5 | Simulated growth.** (**a–c**) Final shapes of the graphene flakes before initiation of the graphene etching. The outlines of the flakes indicate termination by zigzag edges. (**d–f**) Polar plots obtained using the experimental growth rates (corresponding to the six inner most points in the polar plots). (**g–i**) Simulated kinetic Wulff construction of growth.

| Table 1 | Anisotropic growth rates. | | | | | |
|---|---|---|---|---|---|---|
| | **0° (up)** | **60°** | **120°** | **180° (down)** | **240°** | **300°** |
| First layer | 1 | >4.5 | 3.7 | 3.5 | 3.9 | 3.3 |
| Second layer | 1 | 3.3 | 3.5 | 1.9 | 3.7 | 3.9 |
| Third layer | 1 | 3.5 | 4.1 | 1.6 | 3.9 | 4.0 |

The growth rate of graphene zigzag edge as determined from the experiment, plotted as a function of orientation with respect to the up-step direction.

carbon atoms is strongly influenced by changes in the Pt–C interaction at surface steps. There is another small detail hidden in the etching rate of the vacancy island on the flat terrace in Fig. 3j. Instead of being linear, the rate increases with time. This is because the exposed area of Pt, which acts as catalyst for the production of atomic hydrogen, increases with increasing area of the vacancy island.

**STM imaging of graphene edge states.** The structural information derived from the etching kinetics shows stronger interaction between the first layer and the substrate compared with that between stacked graphene adlayers, and a strong coupling of graphene at step edges of the Pt substrate. Here we describe STM imaging that was performed to explore the electronic structure associated with the edges of graphene sheets.

The edges of a graphene island on a flat Pt terrace are shown in the STM image in Fig. 4a, and those of an ALG grown on a SLG in Fig. 4b. In contrast to the depressed edges of the SLG on the Pt surface in Fig. 4a, the ALG edges in Fig. 4b show clearly elevated features. These apparent height variations in the STM images are attributed to differences in the local electronic density of states in graphene. The high electron density in the STM image in Fig. 4b localized at the edges of ALG indicates the presence of pronounced edge states. Such a high electron density at edges of graphene sheets has been predicted by density functional theory (DFT) calculations to occur at hydrogen-terminated zigzag edges[36]. In contrast, the edges of SLG directly in contact with Pt are characterized by depleted electron density, which is attributed to the strong interaction between graphene edge atoms and the flat Pt surface. This observation is in agreement with more recent theoretical descriptions and experimental STM data in the literature, according to which terminal carbon atoms at the graphene edge are either bent down towards the Pt substrate on flat terraces or directly bind to step edges[37,38]. Assuming that this picture holds with increasing domain size, the effect of such strong edge bonding of graphene is to block diffusion, intercalation and transport of reactants and products from and to the growth environment during graphene growth and etching. While it hinders both, growth and etching under the layer, it promotes growth and etching of the topmost layer such as schematically illustrated in Fig. 4d.

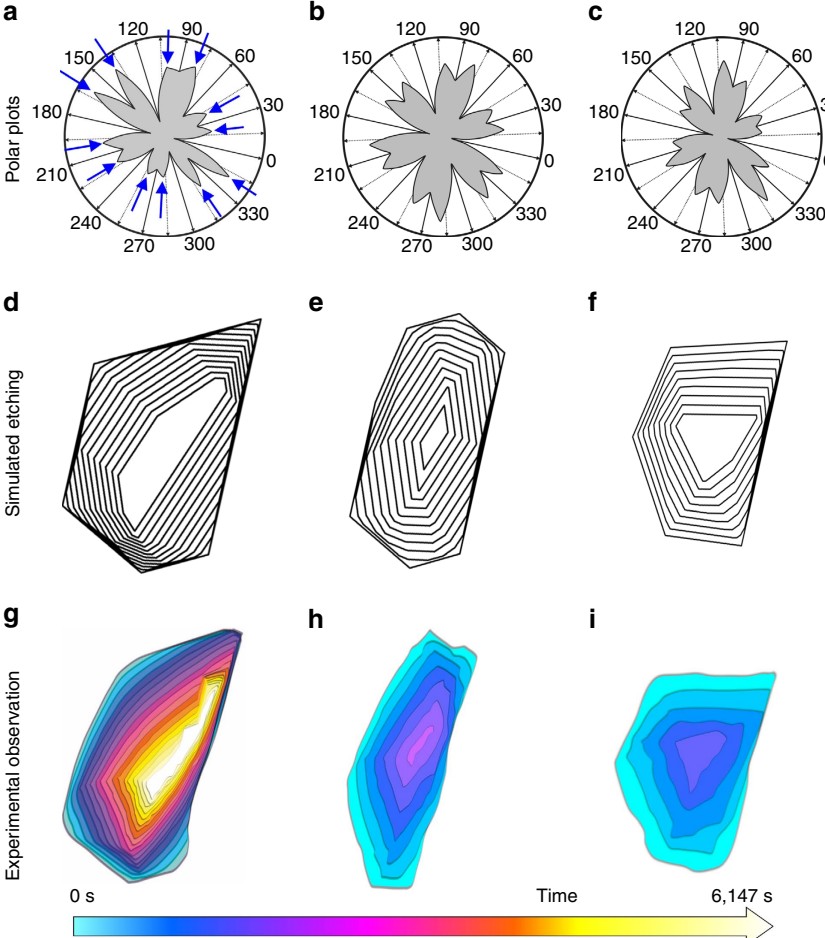

**Figure 6 | Simulated etching.** (**a**–**c**) The same polar plots as in Fig. 5, but this time, the fastest directions indicated by blue arrows in **a** determine the shape evolution. (**d**–**f**) Simulated kinetic Wulff construction of the etching process. (**g**–**i**) Shape evolution of the first, second and third layers during etching as observed in the ESEM.

**Simulation of the shape evolution during graphene etching**. To understand the extent to which the graphene–substrate interactions influence the growth and etching behaviour, we performed simulations of graphene shapes during growth and etching on the basis of experimentally obtained growth rates.

It is known that regular-shaped graphene domains are enclosed by zigzag edges because of their slow growth rate[39–42]. Assuming that all the edges of the observed domains are zigzag terminated, the relative growth rates along different directions were obtained by measuring the distances from the position of the initial nuclei to the respective zigzag edge and dividing it by the growth time (Fig. 5a–c and Table 1). The orientation and the density of steps on the Pt surface breaks up the equal growth rates of the zigzag edges in hexagonally growing graphene into six distinct values[43,44]. The largest difference among the six values exists for the up- and down-step directions. However, compared with the first layer, the difference in the growth rate in up- and down-step direction is less pronounced in the second and third layers. The reduced influence of the substrate steps is a consequence of the different chemical surrounding and corresponding edge state termination of the adlayers. While the edge of the first graphene layer is attached to the catalyst surface by chemical bonding, both the second and third layer edge atoms should be hydrogen-terminated and thus interact only weakly with the graphene layer underneath them.

Using the experimentally obtained growth rates of the six zigzag edges, the growth rates along other directions with different density of kinks were determined under the assumption that growth is controlled by interfacial kinetic processes, using kinetic Wulff construction[39,43,45] (Supplementary Note 2). The resulting polar plots of the orientation dependent growth rates for the first, second and third layers are shown in Fig. 5d–f. Applying the obtained growth rates, the steady-state shapes of graphene domains during growth were simulated, starting with a dodecagon as the nucleus (Fig. 5g–i and Supplementary Note 3). The shapes produced this way are in good agreement with the experimentally observed shapes. Simulations of the etching process were performed simply by reversing the growth process, that is, by using the determined growth rates along the different directions as etching rates Switching from growth to etching induces a shape change of the graphene islands. While the energetically most stable and slowly growing zigzag edges define the shape during growth (inner six points indicated by red arrows in Fig. 5d–f), the shape during etching is determined by the fastest etching ones (indicated by blue arrows in Fig. 6a). These are edges that are tilted with respect to the zigzag direction by 19.1° and are defined by a maximum density of kinks[43]. The shapes produced during simulated growth were used as the starting point for the simulation of etching. Figure 6 shows that the simulation of the etching process is in excellent agreement with the experimentally observed shape evolution during etching. We conclude that such a good agreement justifies a simulation of etching by inversion of growth. The intrinsic growth and etching behaviour of graphene, which is predominantly determined by the most stable edges

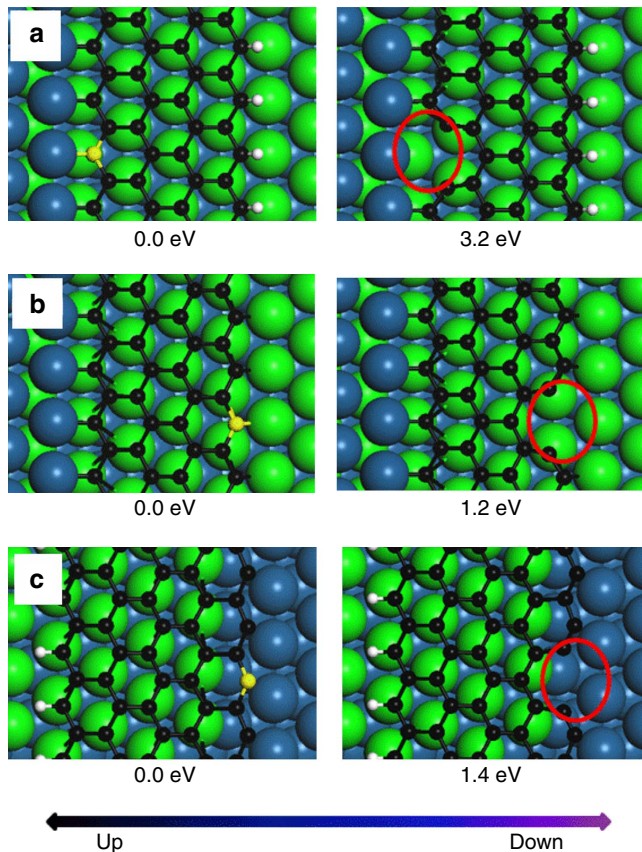

**Figure 7 | DFT models and detachment energies.** Models used for the determination of detachment energies for carbon atoms removed from a zigzag graphene edge that is passivated by a substrate step (**a**), located on the substrate terrace (**b**) and suspended on a substrate step (**c**), respectively. The C and H atoms are represented by black and white spheres, respectively. The Pt atoms are coloured in green and blue to highlight the step structure. The carbon atom that is removed is represented by a yellow sphere and the location of the formed vacancy is highlighted by red ellipses.

during growth and the fastest etching edges during etching is modified by the graphene–Pt coupling to produce the overall rates of carbon attachment and detachment.

To identify the reason for the pronounced difference in step-up and step-down etching that is observed for the first layer, we performed DFT calculations using the models shown in Fig. 7. The energy required for detaching a C atom from the zigzag graphene edge on the Pt substrate is defined as

$$\Delta E = E_A + \varepsilon_G - E_B \qquad (3)$$

where $E_A$ and $E_B$ are the total energy of the structure after and before the detachment of a C atom, respectively, and $\varepsilon_G$ is the energy of a C atom in graphene adsorbed on the Pt (111) surface. From the DFT calculation it follows that down-step etching is slower because it requires breaking a 3.2 eV C–Pt bond compared with the up-step etching that needs to break only a 1.4 eV C–C bond (for more details about the DFT calculation see Methods section). The large difference in step-up and step-down etching speed is thus a consequence of the different bonding types at the step edges.

**Etching behaviour of buried graphene layers.** Up to now we have discussed the stacking sequence in FLG grown by isothermal CVD. However, in the case of Pt, the formation of ALG can also

occur by segregation of dissolved from the polycrystalline Pt foils. We obtained precipitation growth after the termination of an isothermal CVD growth process, during a subsequent cooling step. A typical example for FLG structures that form by segregation during cooling in pure hydrogen atmosphere is shown in Supplementary Movie 6 and illustrated in Fig. 8a–d. The sequence of *in situ* SEM images shows that the size of the lighter grey outer layer decreases with time by hydrogen etching, while the size of the smaller darker patch exhibits no detectable change. This behaviour indicates that the large outer layer is being etched because it is a topmost layer that is directly exposed to hydrogen. In contrast, the small darker patch appears to be effectively sealed off from the hydrogen atmosphere. Indeed, closer inspection reveals that it actually grows due to segregation of C from the Pt as shown in Fig. 8e,f. The second graphene layer corresponding to the small patch therefore grows between the SLG and the Pt substrate by IWC-type stacking[46]. Because of the low solubility of C in solid Pt, which is around 0.0711% at 1,000 °C (ref. 46), these layers are generally limited to small size. The nucleation of a second layer is suppressed further because the insertion of a new layer underneath the SLG requires work against the coupling of the existing SLG to the substrate.

The subtle features in Fig. 8e highlighted by the red and green arrows in vicinity of the dashed green line and magnified in Fig. 8f provide important clues about the mechanism of the C segregation process. Figure 8f reveals that the segregation process occurs at the Pt step edge and involves the gradual sharpening of the corner feature. The surface modification of the Pt is attributed to graphene growth and driven by the Pt–graphene interactions and stabilization of the zigzag edges of graphene[38]. The reconstruction of the Pt step edges involves step bunching through etching and diffusion of the Pt atoms, pushing back the Pt step edges as illustrated by the red arrow, and expansion of the graphene edges marked by the green arrow in the schematic in Fig. 8g. In Supplementary Fig. 4, we show that a wrinkle in the SLG can provide a channel for transporting reactants and products to enable the etching of a buried graphene layer (Supplementary Note 4).

## Discussion

In this work we demonstrate that real-time imaging by ESEM is a versatile and powerful method for the generation of mechanistic insight that is required for the controlled production of FLG with defined number of layers. Direct observation of the shrinking behaviour of individual layers during isothermal etching in pure hydrogen atmosphere provides the missing clue for unravelling the order of layer stacking in FLG growth. In isothermal CVD growth, new layers grow on the topmost layer, while they are inserted between the substrate and already grown layers by C segregation during cooling. The etching rates reveal that the first layer is strongly coupled to the Pt substrate and provides an estimate for the interlayer coupling strength between the second and third layers. By combining theoretical simulations and STM imaging data we show that the coupling of the first layer involves C bonding at the graphene edges, where terminal carbon atoms bend down towards a flat Pt surface or attach laterally to Pt step edges.

The etching of vacancy islands exhibits anisotropy that depends on the density and orientation of Pt step edges. Anisotropic etching serves as further evidence for strong interactions with the Pt surface. We conclude that C bonding at the edges is the decisive factor determining the observed layer stacking. It impedes diffusion, intercalation and transport of reactants and products during graphene growth and etching. It hinders both, underlayer growth and etching, but promotes

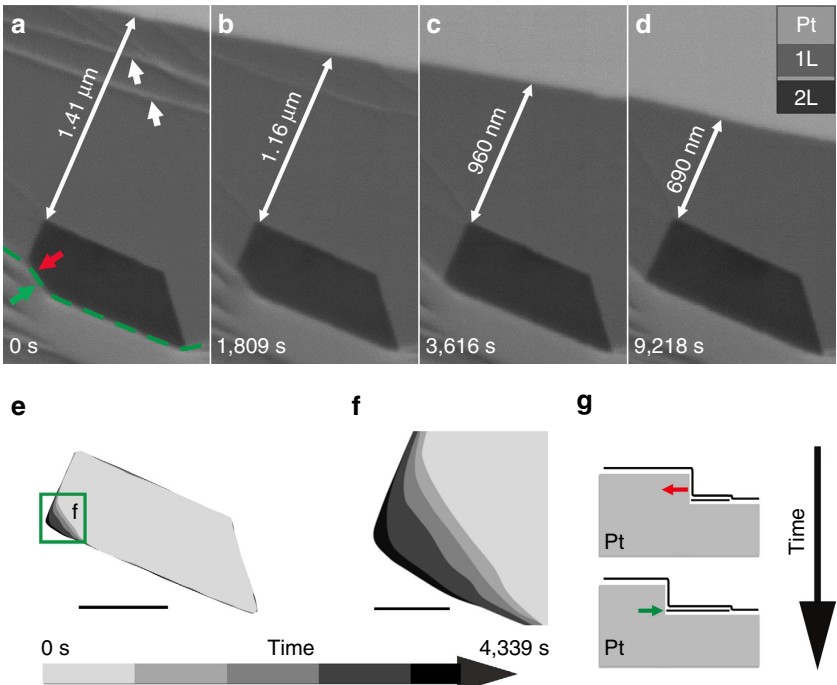

**Figure 8 | Bilayer growth by C segregation during cooling.** (**a**–**d**) Time-lapse image series showing simultaneous etching of the topmost layer and growth of the covered layer underneath. (**e**) Contours of the ALG domain at different times. (**f**) Magnified region of **e**, showing growth by C segregation from a substrate step edge. (**g**) Schematic view showing the retraction of the Pt step edge and growth of a buried carbon layer. Red and green arrows in **a,g** indicate the direction of the Pt step edge movement and C precipitation, respectively. Scale bars in **e,f** measure 500 and 100 nm, respectively.

growth and etching of the topmost layer. Moreover, considering the graphene–metal interaction, we propose that the stacking sequence of FLG during isothermal CVD growth is the same on substrates that are characterized by a similar or stronger graphene–metal interaction and comparable carbon solubility such as Pd, Ru, Ir and Rh. In the case of CVD growth on copper in hydrogen-rich atmospheres, the reversed stacking sequence might be a consequence of hydrogen-terminated edge atoms, as recent DFT calculations have predicted[23]. In this case, diffusion of carbon species into the space between the first layer and the copper substrate and thus, the growth of buried layers is possible. This seems to be in agreement with recently reported statistical analysis of graphene grown on copper[23], according to which no adlayer growth underneath single layers is observed for growth at low hydrogen partial pressure. Under such conditions, the graphene edge atoms might be free to interact with the Cu substrate similarly to the case of Pt. Using simulated growth and etching on the basis of growth rates extracted from the experiment, we were able to demonstrate that under attachment and detachment limited conditions, etching can indeed be treated as the inverse of growth. The agreement between simulation and experimental observation confirms that the shape of graphene domains is predominantly determined by the most stable edges during growth and the edges with the highest kink density during etching. Although scaling up of large-area FLG growth for industrial and commercial processes still faces a number of challenges, this work represents a proof of concept for using *in situ* SEM imaging for developing a real-time feedback loop for controlling coverage uniformity in FLG growth by isothermal CVD. It is possible to grow multilayers, etch some of them and then continue growth of the remaining layers under conditions where no new nucleation events occur and thus, to obtain FLG with a desired number of layers. Finally, this work demonstrates that observation of in-plane dynamics in response to well-controlled experimental environments can provide information about the vertical stacking behaviour of two-dimensional materials and more generally, the capabilities of *in situ* SEM for the study of surface dynamics under controlled environments.

## Methods

***In situ* CVD growth.** *In situ* CVD growth experiments were performed inside the chamber of a commercial ESEM (FEI Quantum 200). The vacuum system of the ESEM was modified and upgraded with oil-free pre-vacuum pumps. The instrument is equipped with a home-made heating stage, a gas supply unit (mass flow controllers from Bronkhorst) and a mass spectrometer (Pfeiffer OmniStar) for the analysis of the chamber atmosphere. The ESEM is not ultra-high-vacuum capable. Owing to the use of rubber O-rings for sealing and the fact that the chamber cannot be baked out, the base pressure of the instrument is around $2 \times 10^{-5}$ Pa, with a residual gas composition mostly comprising water, $N_2$ and $O_2$ (Supplementary Fig. 5). In the ESEM chamber, the oxygen partial pressure is thus below $5 \times 10^{-6}$ Pa. After each sample loading, the chamber was pumped to around $10^{-3}$ Pa, purged with nitrogen and pumped again to $10^{-3}$ Pa successively for several times. Under CVD growth conditions, the pressure is six orders of magnitude higher than the base pressure and constitutes mostly $H_2$ (99.9995% purity) and $C_2H_4$ (99.95% purity). Samples of sizes ranging from $3 \times 3$ to $5 \times 5$ mm were cut from a 0.25 mm-thick polycrystalline Pt foil (99.99% purity) purchased from Alpha Aesar. Before all CVD growth experiments, the chamber of the ESEM was plasma cleaned. The foils were annealed at 1,000 °C under a hydrogen flow of 10 s.c.c.m. at 25 Pa for 1 h inside the chamber. The temperature was measured via a B-type thermocouple that was spot-welded onto the substrate and simultaneously served to ground the sample. CVD growth was performed at 900 °C using a flow of 10 s.c.c.m. $H_2$ and 0.1 s.c.c.m. of $C_2H_4$ at a total chamber pressure of 25 Pa. Hydrogen etching was performed under 10 s.c.c.m. $H_2$ at 900 °C at 25 Pa. During the experiments, the microscope was operated at an acceleration voltage of 5.0–7.5 kV. Images were recorded by a large field detector during CVD growth and etching. No influence of the electron beam on the growth and etching process could be observed. The imaged regions and their respective surroundings showed similar behaviour, as evidenced by changing the magnification or by moving the sample under the beam. Furthermore, no electron beam induced contamination was observed at elevated temperatures.

**Post-growth characterization.** Raman spectroscopy was performed using a Horiba/Jobin-Yvon T64000 spectrometer (Villeneuve D'Ascq, France) with a Coherent Innova 400 (Santa Clara, CA, USA) argon-ion laser operating at 514.5 nm for the excitation. The Raman signal was collected with a multi-channel

charge-coupled device detector. A laser power of 20 mW at the sample and an objective with a × 100 magnification were used. Measurements were performed in confocal mode to reduce the background scattering with respect to the graphene signal. To obtain a satisfactory signal-to-noise ratio, the spectra were recorded with integration times of 60 s, and a total of 10 accumulations.

AFM images were recorded on a Bruker Sharp Nitride Lever probe (SNL-10). Imaging was done in tapping mode using a V-shaped cantilever probe B (silicon-tip on Nitride Lever with frequency $f_0 = 40$–75 kHz and spring constants $k = 0.32$ Nm).

STM measurements were conducted under ultrahigh vacuum in the microscopic chamber at room temperature, with a constant current mode using a home-made W-tip.

High-resolution transmission electron microscopy was performed using an aberration-corrected JEOL ARM electron microscope that is equipped with a cold-field emitter. The image shown in Supplementary Fig. 1 was recorded at an acceleration voltage of 200 kV.

**Computational methods.** All the DFT calculations were carried out by using the Vienna *ab initio* simulation Package[47–49]. The exchange-correlation functional was treated by local density approximation[50]. The projected augmented wave method was used to describe the interaction between valence electrons and ion cores[51]. To calculate the etching of zigzag graphene edges on the Pt substrate with steps, a $4 \times 1$ supercell of the Pt (4 3 3) surface containing three atomic layers was adopted as the substrate, with a zigzag graphene nanoribbon adsorbed on it and the third layer of the Pt substrate fixed during structure optimization. To obtain a commensurate structure, the lattice constants of graphene and Pt are stretched and compressed to their average value. The size of the orthogonal unit cell is $15.26 \times 10.47 \times 30$ Å (ref. 49). The $k$-point grid mesh is sampled by $2 \times 4 \times 1$. The force on each atom is converged to 0.01 eV Å$^{-1}$ during structure optimization, and the energy convergence criterion for the electronic calculation is set to be $10^{-4}$ eV.

**Data availability.** The data that support the findings of this study are available from the corresponding author on request.

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

## Acknowledgements
The work done in Hong Kong PolyU was supported by National Science Foundation of China (21573186 and 21273189). The contribution to this work by G.E. was supported by the U.S. Department of Energy (DOE), Office of Science (OS), Basic Energy Sciences (BES), Materials Sciences and Engineering Division. Y.C. is grateful for support from the NANO-X Workstation in Suzhou and Thousand Young Talents Program in China.

## Author contributions
Z.-J.W and M.-G.W. modified the ESEM, planned and conducted the *in situ* growth and etching experiments, and did most of the ESEM data analysis, TEM and Raman measurements and paper writing; theoretical simulations and implementation of the obtained results was done by J.D. and F.D.; STM measurements were performed by Y.C and Q.F.; important contributions to the interpretation of the results, conception and writing of the manuscript were made by F.D., G.E. and R.S. All authors participated in the scientific discussion.

## Additional information

**Competing financial interests:** The authors declare no competing financial interests.

**How to cite this article**: Wang, Z.-J. *et al.* Stacking sequence and interlayer coupling in few-layer graphene revealed by in situ imaging. *Nat. Commun.* **7,** 13256 doi: 10.1038/ncomms13256 (2016).

