## [Peer Review File · Nature Communications]

Reviewers' comments:

Reviewer #1 (Remarks to the Author):

This paper reports the method of revealing the stacking sequence and interlayer coupling strength in few layer graphene. By combining growth and etching experiments with in situ scanning electron microscopy, the authors succeeded in differentiating between graphene layers forming above or below an already grown layer. They also reproduced shape evolution of graphene islands and vacancy islands by taking into account the interaction of graphene edge atoms with the substrate steps. The observations are novel and the data analyses seem accurate and sound. Overall, the paper is well written and of interesting in the surface science field. However, I doubt that knowing the stacking sequence and interlayer coupling strength in few layer graphene is as important as influencing thinking in the field of surface science or graphene applications. The major issues of graphene growth are enlargement of single crystalline graphene and controlled formation of graphene nanoribbons. Of course, controlled preparation of bilayer and trilayer graphene is also an important subject. If the present work could contribute to such controlled preparation of graphene architectures, and actually achieve some of them, it would influence thinking in the field. Only with knowing the stacking sequence and interlayer coupling strength in few layer graphene, the impact of the paper is relatively small. In this respect, I would not recommend publication of the paper in Nature Communications.

Reviewer #2 (Remarks to the Author):

A. Summary of the key results

This paper gives insight into the physics of graphene growth on Pt.

B. Originality and interest: if not novel, please give references

This is very original. Most researchers do not have these tools

C. Data & methodology: validity of approach, quality of data, quality of presentation

Mostly complete. See suggestions below

D. Appropriate use of statistics and treatment of uncertainties

Not really needed here.

E. Conclusions: robustness, validity, reliability

Good conclusions. Only one difference. Large scale graphene growth will no doubt be done on copper. While the insights here are important, it may not be exactly the same on copper.

F. Suggested improvements: experiments, data for possible revision

See below:

G. References: appropriate credit to previous work?

Reference seem complete

H. Clarity and context: lucidity of abstract/summary, appropriateness of abstract, introduction and conclusions

Overall, well written.

=====

On the whole, a very nice paper. A few additions are in order.
There are a few technical questions that should be answered.
First, a mass spectrometer was mentioned, but no data from this was shown.
Process gases contain small amounts of oxygen (ppm).
The RGA can measure this. Was any oxygen measured, or was it below the measurement threshold?
Second, a little more detail on the mounting and scale of the Pt substrate.
What was the dimension of the foil?
It is not clear what a "home made laser heater" is. How is the Pt substrate thermally isolated and yet electricly grounded?
Third, hot samples emit thermionic electrons. The SEM needs to be protected. FEI sells special adaptors for high temp samples.
What adaptions with respect to high temperature operation were required for this experiment?

Somewhere in the manuscript it should be mentioned that this type of experiment will not be possible on copper.
The vapor pressure of copper is too high and copper vapor would contaminated the SEM. While this should be obvious, some people might not consider this and try these experiments with copper.

Reviewer #3 (Remarks to the Author):

The authors use a real-time imaging technique, which allows for an in situ observation of multilayer graphene growth and oppositely of graphene etching on metal substrate, to provide insight into graphene interlayer coupling and graphene-substrate interaction. This work is original and of fundamental interest. The presentation is clear and most of the arguments are supported by the data provided both in the text and supporting information. However, some points are overlooked, leading to speculative explanations to the etching rate they observed. These critical questions need to be addressed before getting published.

1. The growth mechanism of multilayer graphene grains on Pt is apparently opposite to those previously reported (for example, H. Zhao et al., ACS Nano 8, 10766 (2014)). It has been clearly shown that a graphene adlayer grows underneath the existing layer on a Cu substrate. The authors should discuss the differences.
2. The authors state in the text (Page 4) that the CVD-grown graphene grains are etched by H radicals, which are formed as a result of Pt-assisted dissociation of H₂ molecules. This indicates that the concentration of H radicals is proportional to the exposed Pt areas which are varying in the etching process over time. Then, it is implausible to assume that the etching rate is only a function of temperature and of an apparent kinetic energy, as formulated in Equation 1. It is not surprising to see the three very different etching rates for the grains with different dimensions. The coupling strength between neighbor layers should reach a constant value for the layers far away from the substrate. How are the etching rates and interlayer coupling for these cases?
3. Following the above question, do the authors perform similar experiments at different temperatures to verify their hypothesis? These experiments could provide rich information on the etching dynamics.
4. The three graphene layers shown in the Figure 2 have irregular shapes, indicating a mixed zigzag and armchair edge termination. The interlayer coupling extracted from Equation 1 remains valid for those with hexagonal shapes (mainly zigzag configuration at the edge)?
5. The x-axis label in Fig. 1f and 1g are missing.

Overall, I think this paper is interesting and informative, but still premature for publication if not revised.

Below, I would like to address the points raised by the reviewers and where needed, point out if there are some specific scientific issues that have been overlooked and misconstrued in the reviewing process.

I start with reviewer one, who states that *"The observations are novel and the data analyses seem accurate and sound. Overall, the paper is well written and of interesting in the surface science field."* and

"However, I doubt that knowing the stacking sequence and interlayer coupling strength in few layer graphene is as important as influencing thinking in the field of surface science or graphene applications."

Looking at a paper just recently published in Nature Nanotechnology, titled "Oxygen-activated growth and bandgap tunability of large single-crystal bilayer graphene", doi:10.1038/nnano.2015.322, it is surprising for us that the reviewer does not consider insight in the most relevant growth and stacking mechanisms, as well as graphene-substrate and graphene-graphene interactions as important. Above of that, we also demonstrate that growth and etching can be understood and thus, simulated. Finally, we also reveal the role of surface steps during growth and etching. There is a lot of information in this single paper that cannot be obtained by other methods in such a straight forward and noble manner. This alone would be worth reporting.

Only on the basis of empirical trial and error based growth experiments, graphene research will not move forward. Bandgap engineering is the basis of semiconductor technology and realizing it during controlled growth of few layer graphene is possible only once the underlying growth mechanisms are understood. Probing and determining the interlayer coupling and stacking sequence in real-time during growth provides the ultimate information for controlling bandgap tunability during actual synthesis. The topic of the manuscript combined with the presented *in situ* method is certainly of high relevance not only for the community dealing with two-dimensional growth!

The reviewer writes *"Of course, controlled preparation of bilayer and trilayer graphene is also an important subject. If the present work could contribute to such controlled preparation of graphene architectures, and actually achieve some of them, it would influence thinking in the field. Only with knowing the stacking sequence and interlayer coupling strength in few layer graphene, the impact of the paper is relatively small."*

In the conclusion, we clearly point out that a controlled number of graphene layers can be obtained, once the stacking sequence and interlayer interaction is understood. It is possible to grow multi-layers, etch some of them and then continue growth of the remaining layers under conditions where no new nucleation events occur We have a manuscript under preparation, in which we use selective adlayer etching for controlled SLG growth. If the reviewer requests to see it, we can reveal it, but we do not think that it should be added into this manuscript. In the modified version of the conclusion we have added a more clear statement on how the knowledge of the stacking sequence and

interlayer interaction enables controlled growth of large area single or few layer graphene. Changing the shape and the size of the graphene is achieved as a trivial extension (such as for example lithographic patterning) of the fundamental principles derived from real-time imaging and measurements. If necessary we can provide many real-time movies that richly illustrate these basic facts.

We have added one sentence in the conclusion in order to point out more clearly that the information provided in this paper enables controlled growth of few layer graphene:

“It is possible to grow multi-layers, etch some of them and then continue growth of the remaining layers under conditions where no new nucleation events occur and thus, to obtain FLG with a desired number of layers.”

Reviewer two rates the manuscript as **“This is very original”**, and **“On the whole, a very nice paper”**. He would certainly not reject it and we are very pleased to receive such encouraging comments.

The main concern of reviewer two is: *“Somewhere in the manuscript it should be mentioned that this type of experiment will not be possible on copper.”*

In fact, we have published a paper on direct observation of graphene growth and associated copper substrate dynamics by *in situ* scanning electron microscopy in ACS Nano 9, 1506-1519 (2015) (highly cited paper, ranked in the top 1% of the academic field of Chemistry according to Web of Science). This work was explicitly mentioned and referenced in the manuscript. Clearly, such experiments are possible also on copper.

The reviewer extrapolates further *“Large scale graphene growth will no doubt be done on copper. While the insights here are important, it may not be exactly the same on copper.”*

Indeed, it is not exactly the same on Cu. Graphene growth is even better on Pt as our results that will be published in a manuscript in preparation show. If interested, we can send data from this manuscript, which, although not the subject of the present manuscript, nevertheless becomes important in the light of the reviewers reservation against growth on platinum. (Our response to the rejection took longer because we had to work on the mentioned manuscript in case you want to see it).

Reviewer two made some very good comments and raised questions that helped us to further improve the manuscript:

“First, a mass spectrometer was mentioned, but no data from this was shown. Process gases contain small amounts of oxygen (ppm). The RGA can measure this. Was any oxygen measured, or was it below the measurement threshold?”

The ESEM is not an ultra-high-vacuum capable instrument, mostly due to the use of a rubber O-ring sealing at the chamber door. The base pressure of the instrument is around 2E-5 Pa (2E-7 mbar), with rest-gas constituting of mostly water, N₂ and O₂ (see MS data below). In our chamber, the oxygen partial pressure is thus below 5E-6 Pa. After each sample loading, the chamber was pumped

to around $10\text{E-}3$ Pa, purged with nitrogen up to nearly $1\text{E}3$ Pa, and pumped again to $10\text{E-}3$ Pa successively for 3 times. Under CVD growth conditions, the pressure is six orders of magnitude higher than the base pressure and constitutes mostly H_2 (99.9995% purity) and C_2H_4 (99.95% purity), which were added to the chamber at flows of 10 sccm and 0.1 sccm, respectively. Under growth conditions at 25 Pa, we have to limit the gas flow to the MS chamber with a leak valve, since the latter is operated at pressures in the range of $1\text{E-}6$ to $1\text{E-}4$ Pa. We did not add the MS data recorded during the experiments because we have so far not calibrated it for different pressures. The answer to the reviewers question is: Yes, we can see oxygen also during growth, although its partial pressure is several orders of magnitude below the one of H_2 and C_2H_4 . In control experiments, we have studied the effect of oxygen on the growth and etching, also in combination with *in situ* XPS. We see that some oxygen is beneficial for the growth. We will publish these results as soon as we have finished our experiments. For the moment, the MS mainly serves us to monitor the quality of the vacuum and the growth atmosphere, without providing quantitative numbers about the amount of O_2 and H_2O . That is why we did not add MS data to the manuscript.

In the revised version of the supporting information, we have added more information about the chamber composition and the set-up:

“The ESEM is not an ultra-high-vacuum capable instrument. Due to the use of rubber O-rings for sealing and the fact that the chamber of our instrument can-not be baked out, the base pressure of the instrument is around $2\text{E-}5$ Pa ($2\text{E-}7$ mbar), with a residual gas composition mostly comprising water, N_2 and O_2 (see MS data below). In our chamber, the oxygen partial pressure is thus below $5\text{E-}6$ Pa. After each sample loading, the chamber was pumped to around $10\text{E-}3$ Pa, purged with nitrogen and pumped again to $10\text{E-}3$ Pa successively for several times. Under CVD growth conditions, the pressure is six orders of magnitude higher than the base pressure and constitutes mostly H_2 (99.9995% purity) and C_2H_4 (99.95% purity).”

We have furthermore added a figure with MS data and some descriptive text to the SI:

Residual gas composition in the chamber of the ESEM at a base pressure of $\sim 5 \times 10^{-5}$ Pa shows the presence of mainly water, oxygen, hydrogen and nitrogen. The oxygen signal is higher than the nitrogen signal due to contributions from fragmentation of water by electron impact ionization. Under hydrogen annealing at a chamber pressure of 25 Pa, the gas flow to the MS was restricted by a leak valve to 2.5×10^{-4} Pa. The MS was not calibrated for different pressures and used only to provide qualitative information about the residual gas composition.

“Second, a little more detail on the mounting and scale of the Pt substrate. What was the dimension of the foil? It is not clear what a “home made laser heater” is. How is the Pt substrate thermally isolated and yet electricly grounded?”

Polycrystalline Pt foils of 99.99% purity with a thickness of 0.25 mm were used. The dimension of the samples was in the range between 3x3 to 5x5 mm. We used one of the thermocouple wires to ground the sample, although grounding is not that important in the presence of gases in the atmosphere, which prevent charging of the sample. Our home-made laser heater consists of a commercial infrared laser and a stage that allows direct illumination of the back-side of the sample by the laser (patent pending). We have added the following line to the experimental part in the SI:

Samples of sizes ranging from 3x3 to 5x5 mm were cut from a 0.25 mm thick polycrystalline Pt foil (99.99% purity) purchased from Alpha Aesar.

and:

“The temperature was measured via a B-type thermocouple that was spot-welded onto the substrate and simultaneously served to ground the sample.”

“Third, hot samples emit thermionic electrons. The SEM needs to be protected. FEI sells special adaptors for high temp samples. What adaptations with respect to high temperature operation were required for this experiment?”

So far we have operated the SEM without the shield for many hundreds of hours without any damage to the system. The working distance was generally larger than 10mm. Due to the thermionic electrons, the brightness has to be adjusted and the contrast range gets lower at high temperature. We had to clean the LFD-detector for several times due to the experiments on copper.

Reviewer three states ***“This work is original and of fundamental interest. The presentation is clear and most of the arguments are supported by the data provided both in the text and supporting information.”***

He addresses some important questions that we are happy to address in the following:

“1. The growth mechanism of multilayer graphene grains on Pt is apparently opposite to those previously reported (for example, H. Zhao et al., ACS Nano 8, 10766 (2014)). It has been clearly shown that a graphene adlayer grows underneath the existing layer on a Cu substrate. The authors should discuss the differences.

We mentioned the stacking sequence observed in the case of copper in the introduction and we have cited the mentioned paper at another place. The reviewer is correct in criticizing that we did not further discuss the difference between copper and platinum. We have therefore added the following text and cited a related work in the conclusion:

“In the case of CVD growth on copper in hydrogen rich atmospheres, the reversed stacking sequence might be a consequence of hydrogen terminated edge atoms, as recent DFT calculations have predicted.²³ In this case, diffusion of carbon species into the space between the first layer and the copper substrate and thus, the growth of buried layers is possible. This seems to be in agreement with recently reported statistical analysis of graphene grown on copper,²³ according to which no adlayer growth underneath single-layers is observed for growth at low hydrogen partial pressure. Under such conditions, the graphene edge atoms might be free to interact with the Cu substrate similarly to the case of Pt.”

Added reference: (23) Zhang, X. Y.; Wang, L.; Xin, J.; Yakobson, B. I.; Ding, F. *J. Am. Chem. Soc.* **2014**, *136*, 3040.

“2. The authors state in the text (Page 4) that the CVD-grown graphene grains are etched by H radicals, which are formed as a result of Pt-assisted dissociation of H₂ molecules. This indicates that the concentration of H radicals is proportional to the exposed Pt areas which are varying in the etching process over time. Then, it is implausible to assume that the etching rate is only a function of temperature and of an apparent kinetic energy, as formulated in Equation 1. It is not surprising to

see the three very different etching rates for the grains with different dimensions. The coupling strength between neighbor layers should reach a constant value for the layers far away from the substrate. How are the etching rates and interlayer coupling for these cases?"

The slope of the etching curves show clearly that etching in all layers is detachment limited and thus, not limited by the availability and diffusion of dissociated H₂. Platinum is a very good catalyst for H₂ dissociation and we have observed hydrogen etching up to nearly full coverage of the foil. It remains to be investigated if hydrogen can also be activated directly at the graphene edge. However, the fact that the third layer etches slower is not a consequence of the larger diffusion length for hydrogen species, because the detachment is the rate-limiting step. Otherwise, etching would be diffusion limited. Our simulation of growth and etching, which perfectly reproduce the shape evolution in both directions, further proves that the etching is detachment limited and occurs at the fastest etching edge (the edge that is tilted by 19.16° with respect to the zigzag edge and contains the highest kink density). The etching rates are constant for each layer and not size dependent, instead, the activation energy for detachment is different for different layers. We have a manuscript in which we study the etching behavior of adlayers even for nearly fully covered Pt foils and the results are the same: the etching is detachment limited. If the reviewer likes, we can send him the draft.

"3. Following the above question, do the authors perform similar experiments at different temperatures to verify their hypothesis? These experiments could provide rich information on the etching dynamics."

The reviewer is absolutely right. Temperature dependent etching experiments would provide further insight on the activation energy and etching dynamics. We have performed growth experiments at different temperatures, systematic etching experiments are pending.

"4. The three graphene layers shown in the Figure 2 have irregular shapes, indicating a mixed zigzag and armchair edge termination. The interlayer coupling extracted from Equation 1 remains valid for those with hexagonal shapes (mainly zigzag configuration at the edge)?"

The final shape of the graphene layers just before etching is started, is not easily visible in Figure 5 a-c in the manuscript. Due to the switching of the gases and the changed imaging conditions, we have lost some of the image frames, which is why the simulated etching is only shown for the colored shapes in the case of the third layer. We have added a figure in the supporting information that shows that the edges are zigzag terminated before switching from growth to etching.

Figure SI 3: Final shapes of the graphene flakes before initiation of the graphene etching. The outlines of the flakes indicate termination by zigzag edges. The distance from the center to the respective edges was used to abstract the growth rates in the different directions (the six most inner points in the polar plots, see main text).

“The x-axis label in Fig. 1f and 1g are missing.”

Thank you! It was in Fig. 2f and 2g and is now corrected.

Finally, I would like to point out once more that the manuscript is packed with very relevant information that is needed in order to be able to systematically grow defined numbers of graphene layers. Instead of time consuming and labor intensive trial-and-error optimization approaches, we provide direct insight to 2D growth under relevant conditions and provide data that has so far not been accessible by any other method.

Reviewers' comments:

Reviewer #2 (Remarks to the Author):

First, mea culpa, I missed reference #26.

The authors have made a very good effort to address the concerns of all the referees. I would very much like to see this paper published.

Reviewer #3 (Remarks to the Author):

The authors have clearly addressed most of the questions raised by the three reviewers. The revised manuscript now looks clearer and more convincing.

However, be aware that graphene interlayer coupling is the central issue of the paper. They are responsible for showing a supportive result at another etching temperature, which yields a similar coupling between layers. It is quite acceptable even if the data obtained at a different etching

temperature shows an inconsistent coupling strength due to the varying experimental conditions at a different grain. They cannot bypass this question raised in the first round of review. Providing such supplementary information in the supporting materials will be enough. Overall, this work is technically uneasy and provide the unique experimental perspective on graphene interlayer coupling which becomes more and more important in the rise of other 2D materials. I recommend it for publication in Nature Comm after meeting the request stated above.

Response to Reviewers: Manuscript ID: NCOMMS-16-06096B

First of all, we want to thank the reviewers for their insightful comments and motivating remarks. Once again, the feedback was of high value for us and helped to further improve the manuscript.

In the following, I would like to respond to the reviewers comments:

Reviewer #2:

We were very happy to read that you were satisfied with the additions and clarifications that we made after the first round:

“The authors have made a very good effort to address the concerns of all the referees. I would very much like to see this paper published.”

Thank you for this positive and very motivating feedback!

Reviewer #3:

Thank you for the positive feedback on our modifications that were based on the comments and questions we received in the first round:

“The authors have clearly addressed most of the questions raised by the three reviewers. The revised manuscript now looks clearer and more convincing.”

We do agree with the points that you address:

However, be aware that graphene interlayer coupling is the central issue of the paper. They are responsible for showing a supportive result at another etching temperature, which yields a similar coupling between layers.”

Based on your suggestion, we have conducted additional growth and etching experiments at different temperatures and on different grains. The following material has now been added to the supporting information (including one additional movie, **M3**):

SI 2 Etching rates of 2nd and 3rd layers at 900 degree. **a**, plot of the perimeter versus etching time abstracted from 2nd and 3rd adlayers (see movie M3). Etching was conducted at 900 °C at 25 Pa H₂. The ratio between the etching rates of the 2nd and 3rd layers are similar to the one discussed in the main text (Average of ALG1-4: 1.5 vs. 1.36 in the main text). **b**, corresponding overview image of ALG on a continuous SLG. The ALG stacks from which the shrinking perimeters were recorded and plotted in **a** are indicated by rectangular windows and labelled as ALG1-4. For the analysis, regularly shaped ALG domains were selected in which edges of adlayers were not merged.

Clearly, there is an influence of the Pt structure on the shape of the ALG and thus, on the etching speed. Irrespective of these variations, the 2nd layers always etch at higher rate than the 3rd. The slower etching rate compared to the example discussed in the main text is most likely a consequence of the full coverage of the Pt grain by a single layer graphene. In this case, hydrogen activation by Pt is locally suppressed and etching relies on thermally activated hydrogen.

SI 3 Etching of one ALG stack at different temperatures. **a**, ALG stack showing the 3rd and 2nd layer on a continuous SLG during etching at 925 °C. **b**, shrinking of the respective perimeters during etching at 925 °C, during cooling to 820 °C and at 820 °C. Throughout the observed temperature regime, the etching rate of the 2nd layer was higher than the one of the 3rd layer by a factor of 2.6 at 925°C and 1.3 at 825 °C. **c**, the same ALG stack as shown in **a** showing the 3rd and 2nd layer during etching at 820 °C. Anisotropic shape and etching speed are a consequence of the substrate structure and influence the etching rates. Nevertheless, the 2nd layer etches consistently faster than the 3rd layer.

You wrote that: *“It is quite acceptable even If the data obtained at a different etching temperature shows an inconsistent coupling strength due to the varying experimental conditions at a different grain.”*

Our experiments confirm what is described in main the manuscript at several points, for example in the discussion of the polar plots and the simulated growth and etching. Namely, that the substrate structure and grain orientation has a clear influence on the growth, and correspondingly, on the etching. For this reason, simply taking the shrinking circumference, which is an integral measure, is not accurate enough for precise quantification of the interaction strength. The shape of the adlayers plays a role, which is now clearly stated in the manuscript. The shape of single and adlayer graphene is influenced by the substrate structure during growth and similarly influenced through the etching. Through the additional etching experiments we consistently proof that the interlayer coupling is such that the 2nd layer always etches faster than the 3rd layer, irrespective of the temperature and surface grain orientation. Our conclusions thus remain consistent.

We have added the following text to the discussion of the effect of the interlayer coupling strength on the etching rates (marked in yellow in the main text):

Additional etching experiments revealed that the etching rates depend on the etching temperature and are influenced by the surface structure of the Pt grain (see SI2 and SI3). Hence, anisotropic etching due to irregularities in the morphology of the substrate and resulting anisotropy in the shape of the graphene adlayers should be taken into account. Abstraction of the etching rate based on an integral shrinking perimeter is thus not sufficient for an accurate evaluation of the coupling strength experienced by edge atoms. However, in the case of perfectly flat Pt grains and symmetric hexagonal flakes, etching experiments performed at different temperature should even deliver the corresponding activation energies for etching. Here we refrain from attempting to provide accurate numbers. Instead, we concentrate on the fact that etching experiments performed at different temperatures and on different grains confirmed that the 2nd layer always etches at higher rate than the 3rd layer and that the difference is not related to the size of the etching layers.

We hope that we have now added sufficient evidence for the fact that the strong interaction of the first layer with the Pt substrate in effect weakens the coupling strength between the 1st and 2nd layer, and is responsible for a faster etching of the 2nd layer compared to the 3rd layer. The above mentioned effect of the substrate morphology on the growth and etching speeds are discussed later in the manuscript in the part where we present the simulated growth and etching. In our opinion, everything is pretty consistent now.

You concluded with: *“Overall, this work is technically uneasy and provide the unique experimental perspective on graphene interlayer coupling which becomes more and more important in the rise of other 2D materials. I recommend it for publication in Nature Comm after meeting the request stated above.”*

I hope that you are satisfied with the additions that we made to the supporting information and main text and that you now agree that our manuscript can be published in Nature Communications.

Overall, I want to thank the reviewers again for a very fair and constructive reviewing process. It gave us a chance to further improve our manuscript, enrich it, and back it up by additional experiments. Given the relevant input that we have received, I would be happy to evaluate possibilities for future collaborations once this reviewing process is closed.

With my very best regards,

Marc Willinger